# Autoimmune Rheumatic Disease Flares with Myocarditis Following COVID-19 mRNA Vaccination: A Case-Based Review

**DOI:** 10.3390/vaccines10101772

**Published:** 2022-10-21

**Authors:** Yi Wye Lai, Choon Guan Chua, Xin Rong Lim, Prabath Joseph Francis, Chuanhui Xu, Hwee Siew Howe

**Affiliations:** 1Department of Rheumatology, Allergy and Immunology, Tan Tock Seng Hospital, Singapore 308433, Singapore; 2Department of Cardiology, Tan Tock Seng Hospital, Singapore 308433, Singapore

**Keywords:** COVID-19 mRNA vaccine, myocarditis, post-vaccination complications, rheumatic diseases, vaccination

## Abstract

Since the introduction of coronavirus disease 2019 (COVID-19) messenger ribonucleic acid (mRNA) vaccines, there have been multiple reports of post-vaccination myocarditis (mainly affecting young healthy males). We report on four patients with active autoimmune rheumatic diseases (ARDs) and probable or confirmed myocarditis following COVID-19 mRNA vaccination managed at a tertiary hospital in Singapore; we reviewed the literature on post-COVID-19 mRNA vaccination-related myocarditis and ARD flares. Three patients had existing ARD flares (two had systemic lupus erythematosus (SLE), one had eosinophilic granulomatosis polyangiitis (EGPA)), and one had new-onset EGPA. All patients recovered well after receiving immunosuppressants comprising high-dose glucocorticoids, cyclophosphamide, and rituximab. Thus far, only one case of active SLE with myocarditis has been reported post-COVID-19 mRNA vaccination in the literature. In contrast to isolated post-COVID-19 mRNA vaccination myocarditis, our older-aged patients had myocarditis associated with ARD flares post-COVID-19 vaccination (that occurred after one dose of an mRNA vaccine), associated with other features of ARD flares, and required increased immunosuppression to achieve myocarditis resolution. This case series serves to highlight the differences in clinical and therapeutic aspects in ARD patients, heighten the vigilance of rheumatologists for this development, and encourage the adoption of risk reduction strategies in this vulnerable population.

## 1. Introduction

The ongoing coronavirus disease 2019 (COVID-19) pandemic that started in December 2019 has caused detrimental impacts on healthcare systems, geopolitics, economies, and social norms around the world [1]. Many nations have embarked on mass vaccination campaigns (in addition to employing various safe management measures) using World Health Organization (WHO)-approved vaccines under its emergency use listing (EUL) to confer population immunity [1]. However, there have been multiple reports on the side effects and complications of COVID-19 vaccines [2], including vaccine-related myocarditis and autoimmune rheumatic disease (ARD) flares [1,3]. Notably, a significant rise in post-COVID-19 messenger ribonucleic acid (mRNA) vaccination (Pfizer-BioNTech^®^ or Moderna^®^) myocarditis (mainly affecting males aged 30 years and younger) was reported by the Vaccine Adverse Events Reporting System (VAERS) and in various studies [4,5]. Although myocarditis in older patients with active ARDs following vaccination remains rare, its occurrence in this group of patients can be debilitating and life-threatening [1,6,7]. Herein, we describe the clinical courses of four patients of older age with disease flares or new onset ARD occurring shortly after COVID-19 mRNA vaccination and provide a summary of the current literature on post-COVID-19 vaccine-related myocarditis associated with ARD flares.

## 2. Materials and Methods

The case includes four patients admitted to the Rheumatology, Allergy, and Immunology department of Tan Tock Seng Hospital, a tertiary hospital in Singapore, from March to September 2021, for acute myocarditis and active ARD following COVID-19 mRNA vaccination, were examined. All patients fulfilled the definition of at least probable myocarditis based on the Brighton Collaboration Criteria and the US Communicable Disease Centre (CDC) case definition criteria [8,9]. Three of the four patients had existing ARDs—two with systemic lupus erythematosus (SLE), one with eosinophilic granulomatosis with polyangiitis (EGPA); the fourth had new onset EGPA. The patients fulfilled the Systemic Lupus International Collaborating Clinic’s classification criteria for SLE and ACR/EULAR 2022 classification criteria for EGPA, respectively. A Medline literature search was performed using the MeSH terms “COVID-19 OR SARS-CoV-2) AND (Vaccine OR vaccination) AND (mRNA OR messenger RNA) AND myocarditis AND (rheumatic or autoimmune or rheuma”.

## 3. Results

The key characteristics of each case are described below, with additional relevant clinical information included in Appendix A. Where applicable, the transthoracic echocardiogram (TTE) image is presented in Appendix A while cardiac magnetic resonance imaging (CMRI) images are presented in Appendix A.

### 3.1. Case 1

A 63-year-old Chinese female with existing well-controlled EGPA developed dyspnea one day after her first dose of the COVID-19 mRNA vaccine (Moderna^®^). Two weeks later, her dyspnea worsened and she developed persistent central chest pain for several days before being admitted to the hospital. Her EGPA was diagnosed at the age of 50 after she presented with adult-onset asthma, allergic rhinitis, left sciatic neuropathy, eosinophilia, and positive anti-proteinase three (PR3) antibody tests. Three months before vaccination, the EGPA was assessed as controlled with the patient’s usual dose of azathioprine 50 mg OM, which started eight years prior. The patient had essential hypertension and hyperlipidemia that were well-controlled on long-term lisinopril and simvastatin, respectively, and no history of ischemic heart disease. Upon admission, she was hypoxic (pulse oximetry reading 93% on room air) and the respiratory rate was 18 breaths/minute at rest. The heart sounds were dual and regular; the rate was 90 beats/minute; the jugular venous pressure was not elevated, and there was no peripheral edema. Her breath sounds were reduced at both lung bases, and no rhonchi were heard. Residual reduced sensations of the left foot dorsum and lateral calf, and weak left big toe dorsiflexion (Medical Research Council muscle power grading 4) were present. She had mild eosinophilia of 0.88 (0.00–0.60 × 10^9^/L), a slightly raised C-reactive protein (CRP) of 7.5 (0–5.0 mg/L), and a raised troponin I of 1236 (0–18 ng/L). Tests for anti-PR3 and anti-neutrophil cytoplasmic antibody (ANCA) were negative. Her electrocardiogram (ECG) showed old ischemic changes (Q-wave and T-wave inversions in V2-4). The chest radiograph (CXR) was normal. A transthoracic echocardiogram (TTE) showed a reduced left ventricular ejection fraction (LVEF) of 45% with regional wall motion abnormality (RWMA) in the left anterior descending (LAD) artery territory, characterized by hypokinetic basal to mid-anterior and basal to mid-anteroseptal segments. Although akinesia of the mid-anterior wall and hyperkinesia of the basal and apical anterior walls were seen in an urgent percutaneous coronary angiogram, no signs of coronary artery disease were found. Cardiac magnetic resonance imaging (CMRI) showed marked focal edema, myocardial hyperemia, and a non-ischemic pattern of delayed myocardial enhancement at the apical, anteroseptal, and basal anterior myocardium; the findings are consistent with acute myocarditis.

As the symptoms appeared in close proximity after vaccination, and EGPA was in remission prior, the cause of myocarditis and the EGPA flare was likely due to the COVID-19 mRNA vaccine. Although an endomyocardial biopsy (EMB) would have helped differentiate active EGPA from isolated vaccine-induced myocarditis, it was not pursued because of the risks of this invasive procedure and the results would not influence treatment. The patient improved after pulse methylprednisolone, i.e., 1 g/day for three days, three doses of monthly intravenous cyclophosphamide with normalization of troponin levels, and a normal LVEF of 60% with no RWMA on repeat TTE three months later. mRNA vaccines were subsequently avoided, and she received two doses of a non-mRNA vaccine six months later with no complications.

### 3.2. Case 2

A 64-year-old Chinese male with no pre-existing ARD experienced numbness over the lateral aspect of the left thigh two days after he received his first dose of a COVID-19 mRNA vaccine (Pfizer^®^). He developed persistent purpura over the dorsum of both hands a week later followed by left-sided exertional chest pain two days before his admission. He had anorexia and unintentional weight loss of 5 kg starting two months prior. There was a history of adult-onset asthma (at age 50), sinusitis with nasal polyps, ten years of bilateral sensorineural hearing loss, well-controlled type II diabetes mellitus, and hyperlipidemia. Splinter hemorrhages over the right index finger and non-palpable purpura over both hands were present. Cardiorespiratory, abdominal, and neurological examinations were normal. Initial investigations revealed hypereosinophilia 5.39 (0.00–0.60 × 10^9^/L), normal serum creatinine, a raised erythrocyte sedimentation rate (ESR) of 112 (1–10 mm/h), microscopic hematuria, and a raised troponin I of 1046 (0–18 ng/L). The ECG showed sinus tachycardia, ST-segment depressions, and T-wave inversions in leads V4-V6. RWMA in the right coronary artery (RCA) territory characterized by hypokinetic basal to mid-inferior, basal to mid-inferoseptal and mid-inferolateral segments of the left ventricle, grade 2 left ventricular diastolic dysfunction and multiple intra-cardiac thrombi (up to 15 mm in size), and a normal LVEF of 55–60% were found on the TTE. Blood cultures showed no bacterial growth. Tests for ANCA, anti-myeloperoxidase (MPO), anti-PR3, and anti-phospholipid antibodies were negative. The computed tomography coronary angiogram (CTCA) showed moderately severe stenosis of the mid-LAD and CMRI revealed non-specific myocardial edema and filling defects attributed to small thrombi and vegetations in the left ventricle attached to the papillary muscles. His nerve conduction study was consistent with left lateral femoral cutaneous neuropathy. A skin biopsy showed superficial and deep perivascular infiltrate of eosinophils. The patient was diagnosed with EGPA with endomyocarditis, adult-onset asthma, rhinosinusitis, lateral femoral cutaneous neuropathy, cutaneous vasculitis, and possible glomerulonephritis.

As this patient started experiencing constitutional symptoms and features consistent with EGPA prior to vaccination, it is likely that he had an undiagnosed disease. However, the onset of severe manifestations (chest pain and left thigh numbness) was possibly triggered by the vaccine. Similar to Case 1, an EMB was not pursued due to the risks of the procedure and because it would be unlikely that results would have changed treatment. The patient was treated with pulse intravenous methylprednisolone, 500 mg/day for three days, followed by a tapering dose of glucocorticoids and oral cyclophosphamide, with normalization of the troponin I and resolution of symptoms. A TTE 3 weeks later showed resolution of the cardiac thrombi, improvement of left ventricular diastolic dysfunction (to grade 1), and persistent RWMA in the RCA territory. His dipyridamole stress test three months later showed preserved left ventricular functions with no wall motion abnormality or perfusion defect. The COVID-19 mRNA vaccine was avoided after recovery, and he received two doses of a non-mRNA vaccine eight months later with no complications.

### 3.3. Case 3

A 58-year-old Chinese male with a history of SLE and secondary Sjögren’s syndrome was admitted to the hospital for acute onset of left-sided facial droop and left upper limb weakness occurring three months after receiving the second dose of a COVID-19 mRNA vaccine (Pfizer^®^). Upon further inquiry, he had begun to feel unwell with persistent malaise and generalized myalgia three days after his first dose of mRNA vaccine and had generalized headaches intermittently one month after the second dose (administered three weeks after the first dose). The headaches had increased in frequency and severity, becoming significantly worse three days prior to admission. He had been diagnosed with SLE and secondary Sjögren’s syndrome at age 56, based on disease manifestations of anemia, thrombocytopenia, hypocomplementemia, and positive anti-nuclear (ANA; titer: 1:640, speckled pattern), anti-Ro, anti-La, and anti-Smith antibodies. Anti-dsDNA antibodies were absent. Moreover, 13 months prior to his first mRNA vaccination, he had been assessed as having disease with leucopenia and normocytic normochromic anemia while on prednisolone 8.5 mg/day, and hydroxychloroquine 200 mg/day; but he did not attend the subsequent review. His other medical conditions were hypertension on lisinopril and previously recovered Hepatitis B infection (positive core total antibodies, negative hepatitis B surface antigen).

Upon admission, he was febrile but hemodynamically stable. A physical examination revealed a right gaze deviation, left facial droop, aphasia, anarthria, and petechiae over the flank and lower limbs. A pan-systolic murmur along the lower left sternal edge was heard, and the lungs were clear on auscultation. He had severe hemolytic anemia (hemoglobin (Hb) 4.9 (13.6–16.6 g/dL)) with schistocytes on the peripheral blood film, low haptoglobin, raised lactate dehydrogenase levels, and thrombocytopenia. His coagulation profile was normal. Azotemia, hypocomplementemia, and raised troponin I levels were present. He also had a low ‘a disintegrin and metalloproteinase with a thrombospondin type 1 motif, member 13’ (ADAMS-TS13) level, and positive tests for the ADAMS-TS13 antibody. The TTE showed a reduced ejection fraction of 50%, global hypokinesia, mild circumferential pericardial effusion, and moderate tricuspid regurgitation. A coronary angiogram was not performed in view of the severe anemia with thrombocytopenia; CMRI was not performed due to frequent plasma exchanges. Magnetic resonance imaging of the brain showed multiple scattered supra- and infra-tentorial non-hemorrhagic acute–subacute infarcts with multiple old lacunar infarcts.

Overall, the history and findings suggested active SLE with thrombotic thrombocytopenic purpura and myocarditis. Despite defaulting the follow-up and treatment for his ARD, the patient remained well and symptoms related to this severe SLE flare episode occurred shortly after the mRNA vaccines, suggesting the vaccine was a possible trigger. He was monitored in the high dependency unit, treated with pulse methylprednisolone (total dose 2.5 g) over three days followed by high-dose glucocorticoids, intravenous rituximab, 1 dose of intravenous cyclophosphamide, and 22 plasma exchange sessions. His condition improved with the normalization of troponin I. A TTE five months later showed LVEF of 50% with mild global hypokinesia and a resolution of the pericardial effusion. A post-event dipyridamole stress test revealed a medium partial thickness infarct in the inferior wall extending to the apex and inferolateral wall with no peri-infarct ischemia. The coronary calcium score showed a mild coronary calcification burden with a score of 43 detected in the left main and LAD coronary arteries. Further COVID-19 vaccine doses were not administered.

### 3.4. Case 4

A 43-year-old Burmese female with a known history of SLE was admitted for investigation of a pre-syncopal event, which occurred three weeks after her second COVID-19 mRNA vaccination (Pfizer^®^). One week after the second vaccine dose, she experienced progressive malaise, lethargy, and a bifrontal tension-type headache, followed by exertional dyspnea and reduced effort tolerance in the week preceding admission. She had a 13-year history of SLE with significant manifestations of lupus nephritis and Evans syndrome with positive antinuclear (ANA (titer: >1:640, homogenous pattern), anti-dsDNA, anti-β2 glycoprotein I, and anti-Ro antibodies. There was no history of thrombotic (or bad obstetric) events. Her attending rheumatologist had assessed her SLE to be under control a few days before she received her second vaccine, had reduced prednisolone from 7.5 mg to 6 mg/day, and continued azathioprine 150 mg/day and hydroxychloroquine (285 mg/day). Upon examination, she had pale palmar creases and conjunctival pallor. The cardiorespiratory, abdominal, and neurological examinations were normal. She had severe macrocytic anemia (Hb 5.8 [11.8–14.6 g/dL]); thrombocytopenia (Plt 7 [150–360 × 10^9^/L]); conjugated hyperbilirubinemia, raised reticulocyte counts and LDH, low haptoglobin, hypocomplementemia, and mildly raised troponin I. No schistocytes were seen on the peripheral blood film. The direct Coombs test was positive. Her ECG showed a left bundle branch block and a prolonged QRS with no hyperacute ST-segment change. A chest X-ray (CXR) showed normal lung fields with mild cardiomegaly. The TTE revealed a reduced ejection fraction of 50% and global hypokinesia with no RWMA; findings were consistent with myocarditis.

She was assessed as having active SLE with severe autoimmune hemolytic anemia and myocarditis. As she was on a tapering dose of steroids before she presented, both the mRNA vaccine and immunosuppressive adjustment could have contributed to this SLE flare. She was treated with pulse methylprednisolone (500 mg/day for three days) followed by high-dose glucocorticoids, rituximab, and intravenous immunoglobulin (IVIg). Her effort tolerance improved and troponin I normalized with treatment. A repeat TTE three months later showed improvement in LVEF (to 55%) and a resolution of global hypokinesia. She did not receive further doses of the COVID-19 vaccine.

During the clinical evaluations of these patients, infective etiologies for myocarditis were not detected. None of the cases tested positive for COVID with polymerase chain reaction (PCR) testing.

A literature search performed up until 26 August 2022 yielded 27 articles, of which, only 1 case series of 3 patients described cardiac involvement associated with an ARD flare post-COVID-19 vaccination.

## 4. Discussion

Myocarditis involves the inflammation of the myocardium and can affect the adjacent epicardium and pericardium [10]. Patients present with a wide range of symptoms, including chest pain, dyspnea, cardiogenic shock, and death [10]. Viral infections are the main cause. Other causes include autoimmune diseases, drugs, and vaccines [10,11]. The actual incidence of myocarditis is unknown but is estimated to be 1–10 cases per 100,000 persons annually [10]. Diagnosis is usually made based on clinical presentation and non-invasive imaging [11]. Laboratory findings are non-specific and may include the elevation of troponin and/or CRP levels [11]. EMB remains the gold standard for diagnosis but is not routinely performed due to low sensitivity and potential complications [11]. Hence, the American Heart Association/American College of Cardiology favors the use of CMRI as a diagnostic tool over EMB [11,12,13]. Referencing the Lake Louise criteria, CMRI findings consistent with myocardial inflammation include edema on a T2-weighted study and late gadolinium enhancement on T1 with an increased ratio between myocardial and skeletal muscles (of non-ischemic origin) [14].

Since the introduction of mRNA COVID-19 vaccines, uncommon reports of post-vaccination myocarditis have surfaced, contributing to further concerns over their safety [4]. As of March 2022, more than 18,000 myocarditis and pericarditis events post-COVID-19 mRNA vaccinations in the UK, USA, and EU/EAA population have been submitted to the respective regulatory bodies [5]. Most cases occurred within six weeks of receiving mRNA vaccines, more commonly after the second dose, and predominantly affected males aged 30 and below [15]. There is no existing data that describe myocarditis in ARD patients post-mRNA COVID-19 vaccination or COVID-19 infection. In a recently published nationwide population-based study in Singapore, 25 cases of myocarditis (12 confirmed, 13 probable) were reported against a background of 7,183,889 doses of COVID-19 mRNA vaccines administered [16]. This translates to an overall incidence of 0.35 cases per 100,000 vaccine doses, with the highest incidence in males aged 12–19 years, followed by those aged 20–29 years (3.72 and 0.98 per 100,000 vaccine doses, respectively) [16]. Most cases occurred after the second dose, were mild, and responded well to treatment (details on treatment not available). No myocarditis was reported in females below the age of 40 or for both genders above 60 years of age. The US CDC reported an overall similar rate of myocarditis of 0.35 per 100,000 with the second dose of mRNA COVID-19 vaccine predominantly affecting males aged 18–29 years at 2.43 cases per 100,000 doses administered [17]. An Israeli nationwide study reported about 3 excess myocarditis cases per 100,000 persons following mRNA COVID-19 vaccination that mainly affected younger males (median age 25 years (IQR 20–34), 90.9% males) as observed in other studies [3]. The reported cases were mostly mild and, if required, treated with colchicine, non-steroidal anti-inflammatory drugs, and glucocorticoids; those with left ventricular dysfunction, heart failure, or hemodynamic instability were treated with IVIg along with other cardiac or circulatory support measures [18]. Overall, the prognosis is good as these cases are usually self-limiting [18]. Most patients had resolutions of signs and symptoms with improvements in diagnostic markers and imaging with or without treatment [18].

According to the VAERS databases established in 1990, myocarditis adverse events have also been reported mostly after receiving smallpox, anthrax, typhoid, hepatitis B, and influenza vaccines [15]. However, post-vaccination myocarditis became significantly more prevalent after mRNA COVID-19 vaccines were introduced, and has overtaken smallpox vaccines as the most common cause (more than six times the total number post-smallpox vaccination) [15]. This difference could arguably have resulted from the significantly larger population receiving COVID-19 vaccines as other vaccines are only administered to small vulnerable groups in certain research jobs and travel situations [6]. Notably, some studies have reported frequent occurrences of myocarditis after receiving smallpox and anthrax live vaccines at 59% and 23% of vaccinated individuals, respectively [6]. Hajjo R et al., analyzing the VAERS data, concluded that post-vaccine myocarditis was most frequently reported for live vaccines or vaccines that behave similarly to live vaccines (e.g., mRNA) and possibly viral vector vaccines in comparison with inactivated vaccines [15]. Although the actual mechanism is unknown, postulated mechanisms include the involvement of interferon gamma signaling and T_H_1 immune responses after vaccination, molecular mimicry, PEG-facilitated immune responses, and allergies, as well as the interaction between the SAR-CoV-2 S protein and ACE2 [15].

Myocarditis may be a manifestation of some autoimmune and auto-inflammatory diseases, such as sarcoidosis, Behçet’s disease, EGPA, idiopathic inflammatory myopathy, and SLE. The true incidence is not accurately known but is estimated to be 9–16% of all myocarditis cases, and their incidence and prognoses vary with different ARD diseases [19,20]. Cardiac involvement is observed in 27–47% of EGPA patients and is a predictor of poor long-term prognosis and premature mortality [21]. In SLE, symptomatic myocarditis is reported in 5–10%, although post-mortem prevalence is 40–70%, suggesting that sub-clinical cardiac involvement is common [22,23,24,25]. Whilst ARD flares and new onsets of ARD after COVID-19 vaccinations have infrequently been observed, concurrent myocarditis with active ARD post-vaccination has scarcely been reported [1,7,26]. Our literature review has identified only one case of active SLE in a 31-year-old lady with myopericarditis manifesting as elevated troponin levels and pericardial effusion four days after the first dose of the Pfizer/BioNTech vaccine and treated with an increased dose of steroids [26]. She received a non-mRNA vaccine for her second dose without any adverse effects. Three other cases of pericardial involvement were reported; one SLE patient developed pericardial effusion and possible pericarditis four days after receiving the first dose of the AstraZeneca COVID-19 vaccine, one seronegative inflammatory arthritis patient developed pericarditis seven days after the first dose of the Pfizer/BioNTech vaccine, and one Behçet’s disease patient developed pericarditis seven days after the first dose of the Pfizer/BioNTech vaccine. All three cases improved with increased immunosuppression [1,26]. Hence, ours is the first case series describing the occurrence of concomitant myocarditis with an ARD flare post-COVID-19 mRNA vaccination.

Watad A et al. reported that most patients in their study cohort (17 out of 20) experienced active ARD (both flare and de novo) symptoms after receiving the first dose of a COVID-19 vaccine [1]. A total of 15 subjects experienced a flare of an existing ARD, with more than half having inflammatory arthritis. All except two patients had received mRNA COVID-19 vaccines, the remaining two received AstraZeneca [1]. In a post-market safety assessment of COVID-19 vaccines in 137 patients with ARD, Rotondo C et al. reported 2.2% of patients experiencing ARD flares after one dose of the Pfizer mRNA COVID-19 vaccine [7]. No relapse of ARD was observed after the second dose [7]. In both these studies, the post-vaccination active ARDs were generally mild and patients responded rapidly and well to the escalation of treatment to the ARD. No mortality was reported [1,7]. In our case series, three out of four patients also started experiencing active ARD symptoms after one dose of an mRNA COVID-19 vaccine, similar to the case reported by Patel et al. [26]. Although the numbers in our case series are small, we observed that post-vaccination active ARD with myocarditis occurred earlier after one dose of a vaccine in contrast to after two doses in individuals with isolated post-vaccination myocarditis. Furthermore, isolated post-vaccination myocarditis patients generally require no treatment or only NSAIDs in comparison with our ARD patients, who required aggressive immunosuppressive therapy to treat myocarditis (and other concomitant ARD manifestations, such as thrombotic thrombocytopenic purpura) [20]. The pathogenesis for post-vaccination activation of ARD is unknown but has been postulated to involve an array of innate and adaptive immune mechanisms, including molecular mimicry and adjuvanticity of SARS-CoV-2 vaccines [1]. The adjuvanticity of vaccine mRNA stimulates the innate immune system through endosolic and cytoplasmic nuclear receptors, such as toll-like receptors (TLRs) 3, 7, 8, and 9, and components of the inflammasome, including retinoic acid-inducible gene I and melanoma differentiated-associated gene 5. As a result of altered nucleic acid metabolism and processing, ARDs are postulated to be triggered in susceptible individuals following stimulation of these receptors [1,27,28,29,30].

Patients with ARD have a higher risk of infection and are prone to develop severe infections due to underlying autoimmune dysregulation and treatment-induced immunosuppressive states [7]. As such, vaccinations are particularly recommended for ARD patients to prevent severe infections [31,32]. Despite their rarity, the occurrence of adverse events following immunization still raises public health concerns and could undermine public confidence and trust in vaccination [1]. Thus, the 2019 EULAR recommendations, which promote vaccination during a quiescent disease to reduce the risk of ARD flares (while favoring a good vaccine response), should be adopted [31]. It may also be advisable, therefore, to avoid tapering immunosuppressive treatment when vaccinations are administered, as ARD flares could be potentiated. Non-mRNA COVID-19 vaccines can be considered for ARD patients who experience major ARD flares with myocarditis after an mRNA vaccine. Patients and attending rheumatologists should remain vigilant for symptoms and signs of ARD flares post-COVID-19 vaccinations, particularly with mRNA vaccines, given the possibility of major organ involvement, such as myocarditis, as we have reported. In view of the increased myocarditis risk, ARD patients could consider avoiding strenuous exercises two to three weeks after receiving each dose of an mRNA vaccine. This advice also aligns with the Singapore Ministry of Health post-vaccination guidelines for the general population [33].

## 5. Limitations

The main limitation of this case series is the small number of patients reported from a single tertiary center. This did not allow enough data to calculate the rate of such events within a vaccinated population. Nevertheless, the reported cases serve to highlight the rare occurrences of ARD flares with myocarditis after receiving COVID-19 mRNA vaccines, and alert ARD patients and rheumatologists to this potential adverse effect.

## 6. Conclusions

This case series adds to the expanding knowledge about the complications that can ensue in ARD patients after receiving mRNA vaccines. While isolated post-vaccination myocarditis mainly affects younger males, typically occurring six weeks after two vaccination doses and is usually mild and self-limiting, the patients in our case series who had ARD flares with myocarditis were observed to be older, started experiencing symptoms suggestive of ARD flares/myocarditis after one dose of the vaccine, were more severe, and required aggressive immunosuppression for treatment. This highlights that compared to isolated post-mRNA vaccination myocarditis, post-mRNA vaccination active ARD with myocarditis has different management considerations, including prompt treatment with immunosuppressants for satisfactory outcomes. As knowledge of post-vaccination complications increases with the increasing use of mRNA vaccines, rheumatologists will gain insight and experience into their management, including vigilance for rare complications. Further understanding of the biomechanics of mRNA vaccines and data collected from vaccinated ARD patients would also improve vaccination strategies for this vulnerable population.

## Data Availability

Data are contained within the article or Appendix A.

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
