# Peer review of "Autoimmune Rheumatic Disease Flares with Myocarditis Following COVID-19 mRNA Vaccination: A Case-Based Review"

_vaccines, 2022, doi:10.3390/vaccines10101772_

Round 1
Reviewer 1 Report
I would ask the Authors to put in context and discuss the rate and consenquences of myocarditis by COVID-19 in general population and RMD, as it is the correct comparison group of mRNA-related myocardial disease.
Author Response
I would ask the Authors to put in context and discuss the rate and consequences of myocarditis by COVID-19 in general population and RMD, as it is the correct comparison group of mRNA-related myocardial disease.
Thank you for your comments. Although a direct comparison on the prevalence and consequences of vaccine-related myocarditis between general population and RMD/ autoimmune rheumatic disease (ARD) patients will be ideal, there is currently a paucity of information on the latter group from our literature search. This data is also not routinely collected and reported by the respective vaccine regulators. We have included this point in Line 676-677of the manuscript.
From our literature search, we have observed significantly higher prevalence of myocarditis following SARS-CoV-2 infection compared to post-COVID-19 vaccination in the general population (Relative risk: 15 [95% CI: 11.09–19.81] in the infection group, compared to Relative risk: 2 [95% CI: 1.44–2.65] in the vaccine group).
(reference: https://www.frontiersin.org/articles/10.3389/fcvm.2022.951314/full). We therefore acknowledge the importance of COVID-19 vaccination for all population.
Although we do not know the prevalence of myocarditis in ARD post-mRNA vaccination, our case series highlights potentially severe presentation of myocarditis and ARD flares in this population, which are life-threatening. These patients require urgent aggressive immunosuppressive therapies compared to the general population who generally are managed conservatively or only require NSAIDs. It is therefore important for clinicians to recognise the potentially significant differences in post-mRNA vaccination myocarditis severity in the two population (ARD vs general) such that early intervention with aggressive therapy can be considered.
Reviewer 2 Report
I read with interest the 4 case series where authors try to report a causal association of myocarditis in either denovo or pre-existing rheumatologic cases. Case reports are described well.
Commentss:
1. RMD needs to redefined in the abstract as well as in the manuscript.
2. Authors cited 2 references (Ref 1 and 25): In the first reference cited authors have shown the causal association with mRNA vaccine administration as these cases happened in avg 4 days. The second reference also the time period is compatible with the association. But in the present paper, it is difficult to ascertain if there is an association.
3. Authors reported 2 cases where the flare-up occurred 2wks to 3 months after. It is plausible there could be other triggers!
4. Authors discussed myocarditis in detail after mRNA vaccine, which is now established with hundreds of literature. Authors should describe the possible mechanism of the flare-up or denovo rheumatic diseases.
Author Response
RMD needs to redefined in the abstract as well as in the manuscript.
Thank you for pointing out the missing definition of RMD in the abstract before the abbreviation is used. We have taken this opportunity to amend this term to autoimmune rheumatic disease (ARD) in Lines 11-12 to better reflect the nature/ etiology of rheumatological diseases referred to in our report. The amended terminology/ abbreviation have also been updated throughout the manuscript.
Authors cited 2 references (Ref 1 and 25): In the first reference cited authors have shown the causal association with mRNA vaccine administration as these cases happened in avg 4 days. The second reference also the time period is compatible with the association. But in the present paper, it is difficult to ascertain if there is an association.
Authors reported 2 cases where the flare-up occurred 2wks to 3 months after. It is plausible there could be other triggers!
Addressing both points, the main manuscript has been updated to reflect the temporal sequences of events more clearly. Some of the details were previously found only in the attached Table 2 and have been transcribed to the updated manuscript in Line 116-118, 293-296,409-413,454-455. In the attached revised Table 2, we have also provided further timeline from vaccination to onset of both ARD and cardio-respiratory symptoms (that may suggest myocarditis). The onset of cardio-respiratory symptoms following any (either first or second dose) mRNA vaccination was within a week for two cases (Case 1 and 2), at around 2 weeks for Case 4, and 3-4 weeks for Case 3. The onset of all-inclusive ARD flare symptoms for all cases cited in our case series was within a week from receiving any mRNA vaccine. Given the close proximity of vaccination and onset of any ARD flare symptoms, we identified COVID-19 mRNA vaccine as potential trigger of ARD flare with myocarditis in our cohort.
Authors discussed myocarditis in detail after mRNA vaccine, which is now established with hundreds of literature. Authors should describe the possible mechanism of the flare-up or denovo rheumatic diseases.
We have expanded upon the original explanation and included more details in Lines 796-775. The corresponding references have also been updated.
Reviewer 3 Report
This manuscript reports four patients with active RMD and probable or confirmed myocarditis following COVID-19 mRNA vaccination managed at a tertiary hospital in Singapore and reviewed the literature on COVID-19 mRNA vaccine-related myocarditis and RMD flares post mRNA vaccination.
Comments:
1. The description of 4 cases is interesting, although it is a very limited series that probably does not allow the findings of these 4 cases to be extrapolated to the group of patients receiving COVID-19 mRNA vaccines. For this reason, the following paragraph of the Conclusion section seems insufficiently supported: "While isolated post vaccination myocarditis mainly affects younger males, typically occur six weeks after 2 vaccination doses and is usually mild and self-limiting, in our case series of older aged patients, myocarditis associated with RMD flares post COVID-19 vaccination occurred after one dose of vaccine, were associated with other features". This sentence indicates an extrapolation rather than a direct conclusion.I2. It may not be possible to establish definitively the time that vigorous physical activity should be limited. Why only 2-3 weeks as indicated in the last sentence of the Discussion section? Perhaps it would be better to indicate that this period is discretionary and may vary according to the clinical situation of each patient.
3. No table is provided in the manuscript to facilitate a summary of the results, such as the 4 cases described here. Likewise, it would be useful to provide a summary table with data from some of the main case series.
Author Response
The description of 4 cases is interesting, although it is a very limited series that probably does not allow the findings of these 4 cases to be extrapolated to the group of patients receiving COVID-19 mRNA vaccines. For this reason, the following paragraph of the Conclusion section seems insufficiently supported: "While isolated post vaccination myocarditis mainly affects younger males, typically occur six weeks after 2 vaccination doses and is usually mild and self-limiting, in our case series of older aged patients, myocarditis associated with RMD flares post COVID-19 vaccination occurred after one dose of vaccine, were associated with other features". This sentence indicates an extrapolation rather than a direct conclusion.
Thank you for your comments. We recognise the limitation of our series and the conclusion that we can draw or infer from the data we have. This has been highlighted in the “Limitation” section of our manuscript. We have also taken your comments into consideration and rephrased the corresponding sentence in your comments as “While isolated post-vaccination myocarditis mainly affects younger males, typically occurring six weeks after 2 vaccination doses and is usually mild and self-limiting, the patients in our case series who had ARD flares with myocarditis were observed to be older, started experiencing symptoms suggestive of ARD flares/ myocarditis after one dose of vaccine, were more severe and required aggressive immunosuppression for treatment.” (see Line 852-857).
It may not be possible to establish definitively the time that vigorous physical activity should be limited. Why only 2-3 weeks as indicated in the last sentence of the Discussion section?
Round 2
Reviewer 2 Report
Authors have adequately revised my querries.